# Distributed Vibration Monitoring System for 10 kV-400 kVA 3D Wound Core Transformer under Progressive Short-Circuit Impulses

**DOI:** 10.3390/s24134062

**Published:** 2024-06-21

**Authors:** Jiagui Tao, Sicong Zhang, Jianzhuo Dai, Jinwei Zhu, Heng Zhao

**Affiliations:** Electrical Power Academy of Sciences, State Grid of Jiangsu Electric Power Co., Ltd., Nanjing 210024, China; zhangsc@js.sgcc.com (S.Z.); daijz@js.sgcc.com.cn (J.D.); zhujw9@js.sgcc.com.cn (J.Z.); zhaoh10@js.sgcc.com.cn (H.Z.)

**Keywords:** 3D wound core transformer, windings deformation, progressive short-circuit impulse, wireless monitoring system, wireless sensor node

## Abstract

As large-scale, high-proportion, and efficient distribution transformers surge into the grids, anti-short circuit capability testing of transformer windings in efficient distribution seems necessary and prominent. To deeply explore the influence of progressively short-circuit shock impulses on the core winding deformation of efficient power transformers, a finite element theoretical model was built by referring to a three-phase three-winding 3D wound core transformer with a model of S20-MRL-400/10-NX2. The distributions of internal equivalent force and total deformation of the 3D wound core transformer along different paths under progressively short-circuit shock impulses varying from 60% to 120% were investigated. Results show that the equivalent stress and total deformation change rate reach their maximum as the short-circuit current increases from 60% to 80%, and the maximum and average variation rate for the equivalent stress reach 177.75% and 177.43%, while the maximum and average variation rate for the total deformation corresponds to 178.30% and 177.45%, respectively. Meanwhile, the maximum equivalent stress and maximum total deformation reach 29.81 MPa and 38.70 μm, respectively, as the applied short-circuit current increased to 120%. In light of the above observations, the optimization and deployment of wireless sensor nodes was suggested. Therefore, a distributed monitoring system was developed for acquiring the vibration status of the windings in a 3D wound core transformer, which is a beneficial supplement to the traditional short-circuit reactance detection methods for an efficient grid access spot-check of distribution transformers.

## 1. Introduction

With the rapid development of China’s economy and the guidance and promotion of policies related to the 2021–2023 Energy Efficiency Improvement Plan for Distribution Power Transformers [1], the demands for the investigation and deployment of efficient transformers are continuously increasing. Since the 13th Five-Year Plan, the proportion of the newly invested efficient transformers has increased from 12% in 2015 to 46%, which puts higher requirements on the comprehensive performance, especially their anti-short circuit capabilities [2]. During the operation of transformers, extreme operating conditions of high current shock impulse will occur occasionally, and the deformation of windings under the impulse electromagnetic force is the direct cause of insulation rupture, inter-turn short circuits, and other unexpected accidents [3,4]. For the damaged transformers withstanding fault impulses, windings made of metal materials mostly exhibit typical plastic deformation characteristics. Under most working conditions, the inconspicuous elastic deformation of the winding will recover to its initial state once the impulse is removed. However, under short-circuit shock impulse conditions, plastic deformation of the winding continuously increases with the applied short-circuit current. In the event that the plastic deformation of the transformer winding accumulates a certain value, the winding of the transformer will be damaged [5]. The stress and plastic deformation of windings induced by varying and alternating electromagnetic forces are pivotal indicators for the design and testing of distribution transformers.

The winding deformation will occur once the anti-short circuit capability of a transformer is insufficient, and then directly putting it into operation may cause safety hazards. Therefore, whether the mechanical state of the winding can be effectively detected is an effective indicator for judging the results of sudden short-circuit experiments. Initially, related research mainly focused on the establishment of theoretical models for windings. Madin and Watts et al. derived the mechanical motion equivalent equation in 33.3 MVA single-phase large power transformers during short circuits, and the axial force and coil displacement were estimated to further improve the dynamic response of transformers under short circuit conditions [6,7]. Behjat et al. established a time-stepping finite element model for power transformers, which was combined with external circuit equations to predict the dynamic response of transformers under actual power supply and external load [8]. In addition, researchers reported some related work on steady-state modeling and transformer analysis. Taking a 10 kV/400 V transformer as an example, Ji et al. established an equivalent mathematical model for the axial vibration of the winding based on the actual structural characteristics of the power transformer winding and the mechanical properties of the insulation pad. Efforts were devoted to revealing the characteristics of the axial vibration of the winding under stable operating conditions and analyzing the relationship between the magnitude of the clamping force, the natural vibration frequency of the winding, and the acceleration signal of the winding vibration [9,10]. Wang et al. derived a calculation model for generated residual stress during the winding process of circular ring windings, and the radial short circuit strength of the inner winding was simulated using an experimental transformer taking into account the influence of the residual stress [11]. Yu et al. proposed the axial vibration mechanism of transformer windings based on the mass–spring model and built a vibration testing system for a single winding model of a transformer using a piezoelectric accelerometer. The distribution of axial vibrations of transformer windings, as well as the influences of load current fluctuations and winding looseness, was explored [12]. Zhang et al. selected the SZ-50000/110 kV transformer as an example to establish a nonlinear vibration model of the winding axis and a calculation method was proposed for the dynamic compression force of the coil, pad, and pressure plate, as well as the axial bending stress of the conductor [13,14,15]. Liu et al. employed an SFSZ8-40000/110 transformer to establish a “mass–spring–damping” model for axial vibration of windings in consideration of pressure plates and coils, and a prediction method was proposed for equivalent mass, stiffness, and natural vibration characteristics [16,17]. Yang et al. proposed a method combining optimized adjustable quality factor wavelet transform (TQWT) and Laplace feature mapping (LE) to analyze the vibration signal of transformers under short-circuit impulse, achieving accurate detection of the transformer winding state under short-circuit impulse [18]. On the basis of the increasingly mature establishment and theoretical calculation of transformer winding deformation models, research on the design and development of transformer winding deformation monitoring devices/systems has emerged [19,20,21]. As a passive dielectric sensor element, Melo et al. employed Fiber Bragg Grating (FBG) pressure sensors to deploy on the windings of the transformer for monitoring the static and dynamic pressure in high-voltage winding transformers during events such as short-circuits and inrush currents. Experimental results show that the static sensitivity of 0.911 pm/N was obtained by polyether ether ketone (PEEK) sensors within the range of 800 N–1500 N, which is approximately 4.47 higher than that of the transformer board (TB)-based counterparts [22]. Shi et al. proposed a two-wire model of the winding structure by the principle of least action to describe the dynamic winding behaviors concisely. Then, the acoustic signal from a 110 kV transformer was captured on a multiple short-circuit test platform, and the constructed theory was also well validated [23]. Liao et al. designed a vibration acquisition system for the distributed transformer, which converts the sampled vibration signals into digital signals to the control unit. Then, the health status of the windings is obtained through computer analysis of the vibration signals [24]. Cao et al. proposed a comprehensive monitoring method for transformer winding deformation status and developed a multi-information monitoring system for winding deformation. Meanwhile, mathematical equations and solving models were established to explore the correlation between vibration and reactance information by winding deformation. Loose windings and deformed faulty windings can be effectively monitored to provide a basis for the application of multi-information monitoring and digital twin technology [25]. Xu et al. proposed the deployment principles of sensor nodes by referring to experimental data on the selection of measurement points for transformer vibration detection [26,27,28,29].

In summary, previously focused efforts on the anti-short circuit capability of transformer windings are mainly dedicated to the theoretical modeling of stacked iron core distribution transformers and developing vibration monitoring systems. However, with the upgrading of traditional power transformers and the high proportion of efficient transformers entering the grid, these types of efficient distribution transformers are also affected by short-circuit shocks in engineering practice and often suffer from multiple shocks. Nevertheless, present research on the anti-short circuit capability for efficient 3D wound core transformers is still lacking, and the testing standards and empirical data for existing traditional stacked iron core transformers are no longer applicable. Therefore, in this work, a theoretical model was built for a 3D wound core transformer with reference to the specific model of S20-MRL-400/10-NX2. The influence of applied progressive short-circuit shock impulse on the stress and strain distribution of the transformer windings was explored. In addition, a scheme of optimum deployment for the wireless sensor nodes was also provided in light of the obtained results, and a distributed system for vibration monitoring of the 3D wound core transformers during short circuits was also developed. Compared with previously reported counterparts, major contributions could be summarized as follows: (i) Winding deformation characteristics of a 10 kV–400 kV 3D wound core distribution transformer were systemically studied under progressive short-circuit impulses, providing a theoretical reference for the deployment of the distributed sensors. (ii) A distributed acceleration sensor monitoring system has been constructed, which is a beneficial supplement to the traditional reactance measurement method and further enriches the methods for efficient grid access permit testing. (iii) A quality evaluation method for a 3D wound core distribution transformer was proposed under the newly released energy efficiency standards.

## 2. 3D Simulation Model Construction and Data Extraction Path Determination

The simulation model of the target transformer was established with a real 3D wound core transformer (model S20-MRL-400/10-NX2) as a reference, as shown in Figure 1. In view of the relative complexities of the internal structure, the material parameters of the iron core and winding as well as the connection between the main components of the 3D wound core distribution transformer and the key parameters for model construction including the M&H hysteresis loop of ferromagnetic materials and the conductivity of the windings are determined and listed in Table 1. To accurately estimate the parameters such as short-circuit current, leakage magnetic field, and strain of transformers, lamination processing and manufacturing methods by using toroidal stacked amorphous ribbons should be considered. For the 3D model establishment, firstly, the three-dimensional x-y-z coordinate system was determined, and then the multiple stacked sectional shapes of iron cores in the x-z plane were generated based on the actual height and length of the iron core. According to the thickness of the stacked layers, multiple stacked profile shapes will be established for gradient stretching along the y-axis direction, and each group of stacked layers will be combined into a single-frame iron core. Secondly, the high/low-voltage winding model was established by combining the three frame columns of the iron core model based on the actual size of the ABC three-phase high/low-voltage windings. The axis of the iron core three-phase core column on the x-y geometric coordinate plane was selected as the center of the circle of the three-phase high and low-voltage windings, and then a strictly concentric profile for the high/low-voltage windings in consideration of the inner and outer radius, namely, the high-voltage primary winding for the outer side, and the low-voltage secondary winding for inner side. Finally, the cross-sectional profile of the three-phase winding along the z-axis direction was stretched according to the real size to generate a geometric model of three-phase high/low-voltage windings.

To explore the distributions of equivalent stress and total deformation from the core winding of a 3D wound core transformer under progressive short-circuit impulse currents shock impulse, three paths of *l*_1_, *l*_2,_ and *l*_3_ were selected from high/low-voltage sides of the windings to extract the distribution data of the internal equivalent stress and total deformation for comparison. In view of the three-axis symmetrical structure of the three-phase 3D wound core transformer, the outer line with the farthest distance between the A-phase winding and the other two was selected as path *l*_1_. The intersection line between the *l*_1_ symmetrical plane and A-phase winding was selected as path *l*_2_, while the intersection line between the perpendicular orthogonal plane passing through the center of *l*_1_ and the A-phase winding was selected as path *l*_3_, as illustrated in Figure 1a. Figure 1b shows the specific positions of *l*_1_ and *l*_2_ along high/low-voltage windings in the cross-sectional view of the A-phase of the 3D wound core transformer, and the data were extracted from the outer side of the A-phase winding. The distribution characteristics of the elastic–plastic deformation of the transformer cores under progressive short-circuit current excitations were investigated by comparing simulation results of the equivalent stress and total deformation from high/low-voltage windings along the corresponding three paths.

## 3. Elastic–Plastic Deformation of the Single-Phase Winding under Progressive Impulse

Vibration from the windings is one of the main sources during the on-line operating transformers. Conventionally, the windings will be moved by electrodynamic force generated from the leakage magnetic field, and the alternating characteristics of this field have a periodic force on the windings, ultimately leading to periodic vibrations of the windings. In light of this, elastic–plastic deformation of the windings for Phase A was investigated under progressive short-circuit impulses varying from 60% to 120% of the short-circuit current. Figure 2 shows the equivalent stress distribution cloud map and the equivalent stress distribution along paths *l*_1_, *l*_2_, and *l*_3_ in the three-phase 3D wound core transformer under the application of short-circuit electrodynamic force induced by an 80% short-circuit current impulse. From Figure 2a,b, high/low-voltage windings bore uniform equivalent stress under an 80% short-circuit current impulse. The maximum equivalent stress reached 13.24 MPa, which is 77.72% higher than the maximum equivalent stress under the excitation of a 60% short-circuit current. Figure 2c shows the equivalent stress distribution data of high/low-voltage windings along paths *l*_1_, *l*_2_, and *l*_3_ under an 80% short-circuit current impulse. We noticed that the distribution variation tendency of equivalent stress along different paths in the axial direction is basically the same, and the stress in the middle of the windings is relatively higher than in other positions. The maximum equivalent stresses along the high-voltage winding paths *l*_1_, *l*_2_, and *l*_3_ are 6.92 MPa, 10.16 MPa, and 9.19 MPa, respectively. The maximum equivalent stresses along the low-voltage winding paths *l*_1_, *l*_2_, and *l*_3_ are 8.68 MPa, 6.50 MPa, and 6.76 MPa, respectively. Compared with the equivalent stress with the corresponding path under a 60% short-circuit current impulse, the difference in equivalent stress between each path increases. For example, the difference in stress between high-voltage winding paths *l*_2_ and *l*_1_ increased from 1.75 MPa to 3.24 MPa, while the difference between low-voltage winding paths *l*_2_ and *l*_1_ increased from 1.37 MPa to 2.18 MPa, corresponding to the increments of 1.49 MPa and 0.81 MPa, respectively. Results indicated that the stress increase at the position—where the original equivalent stress is greater—is more significant under the same short-circuit current impulse.

Meanwhile, under 80% short-circuit current impulse, the deformation distribution cloud map and the deformation distribution along paths *l*_1_, *l*_2_, and *l*_3_ of the 3D wound transformer subjected to short-circuit electrodynamic force are shown in Figure 3. From Figure 3a,b, uniform deformation generated by the high/low-voltage windings increased by an 80% short-circuit current impulse, then the maximum deformation reached 17.26 μm, exhibiting 78.31% enhancement relative to the maximum deformation under the excitation of a 60% short-circuit current. Figure 3c shows the deformation distribution of high/low-voltage windings along paths *l*_1_, *l*_2_, and *l*_3_ under an 80% short-circuit current impulse. The observable deformation distribution trend along different paths in the axial direction is essentially identical, with a different highest amplitude occurring around the central region. The maximum total deformations along the high-voltage winding path of *l*_1_, *l*_2_, and *l*_3_ are 10.61 μm, 16.11 μm, and 15.56 μm, respectively. Correspondingly, the maximum total deformations along the low-voltage winding path of *l*_1_, *l*_2_, and *l*_3_ are 10.10 μm, 5.85 μm, and 8.54 μm, respectively. Compared with the total deformation under 60% short-circuit current excitations, the discrepancy in the total deformation from each path increases. For instance, the difference between high-voltage winding paths of *l*_2_ and *l*_1_ increased from 2.86 μm to 5.50 μm, while the difference between low-voltage winding paths *l*_1_ and *l*_2_ increased from 2.53 μm to 4.25 μm. The results of the increase in total deformation indicate that the significant deformation increases at the location where the original total deformation is higher under the same short-circuit current impulses.

Progressively increasing the impulse short-circuit current intensity to 120%, the equivalent stress distribution cloud chart and the equivalent stress distribution along the paths of *l*_1_, *l*_2_, and *l*_3_ of the three-phase windings around the iron core under short-circuit electrodynamic force are shown in Figure 4. Under the excitation of a 120% short-circuit current, the uneven equivalent (Von Mises) stress born by the high/low-voltage windings of the iron core increases significantly. The maximum equivalent stress reached 29.81 MPa, which is 300.13%, 125.15%, and 43.73% higher than that under 60%, 80%, and 100% short-circuit current excitations, respectively. Figure 4c shows the equivalent stress distribution of high/low-voltage windings along the paths of *l*_1_, *l*_2_, and *l*_3_ under 120% short-circuit current excitation. The maximum equivalent stresses along the high-voltage winding paths of *l*_1_, *l*_2_, and *l*_3_ are 15.67 MPa, 22.79 MPa, and 20.87 MPa, respectively, while the maximum equivalent stresses along the low-voltage winding paths of *l*_1_, *l*_2_, and *l*_3_ are 19.91 MPa, 15.51 MPa, and15.49 MPa, respectively. Compared with the equivalent stress of the corresponding path under 60%, 80%, and 100% short-circuit current excitations, the difference in equivalent stress from each path increases correspondingly. For example, the difference between high-voltage winding paths of *l*_2_ and *l*_1_ is 7.12 MPa, which increases by 4.59 MPa, 2.47 MPa, and 2.06 MPa, respectively. By contrast, the difference between low-voltage winding paths of *l*_2_ and *l*_1_ is 4.40 MPa, which increases by 3.03 MPa, 2.22 MPa, and 0.14 MPa, respectively. The results of the increase in equivalent stress indicate that the equivalent stress received by the winding increased with the short-circuit current, but the growth rate slowed down. Therefore, the acceleration of the increase in equivalent stress on the transformer winding is relatively high in the initial state of short-circuit excitations.

The total deformation distribution cloud chart of the three-phase winding and the total deformation distribution along the paths of *l*_1_, *l*_2_, and *l*_3_ of the iron core under the short-circuit electrodynamic force from 120% short-circuit current excitations are summarized, as illustrated in Figure 5. Under the excitation of a 120% short-circuit current, the uneven deformation generated by the high/low-voltage windings of the coil core transformer continuously increases. The maximum total deformation reached 38.70 μm, which is 299.79%, 124.22%, and 43.28% higher than that under 60%, 80%, and 100% short-circuit current excitations, respectively. Figure 5c shows the total deformation distribution of high/low-voltage windings along the paths of *l*_1_, *l*_2_, and *l*_3_ under a 120% short-circuit current impulse. The maximum total deformation along the high-voltage winding path of *l*_1_, *l*_2_, and *l*_3_ are 24.18 μm, 36.17 μm, and 35.32 μm, while the maximum total deformation along the low-voltage winding path of *l*_1_, *l*_2_, and *l*_3_ are 23.66 μm, 13.20 μm, and 19.17 μm, respectively. Compared with the total deformation of the corresponding path under 60%, 80%, and 100% short-circuit current excitations, the difference in total deformation between each path progressively increases. For example, the difference between the high-voltage winding paths of *l*_1_ and *l*_2_ is 11.99 μm, and the difference between the low-voltage winding paths of *l*_1_ and *l*_2_ is 10.46 μm.

Figure 6 shows the equivalent stress and the corresponding variation rate of the iron core windings under progressively short-circuit current excitations with intensities varied from 60% to 120%. Figure 6a–c shows maximum, minimum, and average equivalent stresses and variation tendency under progressively short-circuit current excitations. The equivalent stress on the winding is approximately proportional to the amplitude of the short-circuit current, but the variation rate exhibits different behaviors. The growth rate of equivalent stress is faster than that of the excitation short-circuit current. For example, once the short-circuit current increases from 60% to 80%, the current change rate is 133.33%, and the corresponding maximum, minimum, and average equivalent stresses increase from 7.45 MPa, 0.28 MPa, and 2.07 MPa to 13.24 MPa, 0.51 MPa, and 3.68 MPa, corresponding to the variation rate of 177.75%, 180.94%, and 177.43%, respectively. We infer from the results that (i) the variation rate of the minimum equivalent stress is evident, and the variation rate of the maximum and average equivalent stresses is essentially identical; (ii) once the short-circuit current impulse exceeds 100%, the growth rate of the equivalent stress on the winding gradually slows down as the short-circuit current increases; and (iii) the increment of the maximum equivalent stress reaches its maximum with the increase of the short-circuit impulse current. Therefore, as the short-circuit impulse current increases, the stress situation of the coil at the location where the equivalent stress is greater will become more severe, and the location where the winding is subjected to the maximum force should be the focus of quality inspection for coil core transformers.

Figure 7 shows the total deformation and the corresponding variation rate of the iron core windings under progressively short-circuit current excitations with intensities varied from 60% to 120%. Figure 7a,b shows the trend and rate of change of the maximum and average total deformation with different short-circuit currents, respectively. The total deformation of the winding is approximately proportional to the amplitude of the short-circuit current, and the deformation growth rate is faster than the growth rate of the excitation short-circuit current. For example, once the short-circuit current increases from 80% to 100%, the variation rate of the applied current is 125%, and the corresponding maximum and average total deformation are from 17.26 μm and 3.66 μm to 27.01 μm and 5.73 μm, respectively, with the variation rates of 156.47% and 156.48%. We infer from the results that (i) the maximum variation rate is in accordance with the average total deformation; (ii) Once the short-circuit impulse current exceeds 80%, the total deformation growth rate of the winding gradually slows down as the short-circuit current increases; and (iii) the maximum deformation increment occurs at the point of maximum deformation as the short-circuit impulse current increases. Therefore, as the short-circuit impulse current increases progressively from 60% to 120%, the deformation of the coil in the area where the original deformation occurred will intensify faster, and the distributed sensor should be deployed in the area where the deformation occurs significantly, providing a guidance for the on-site deployment of wireless sensor nodes.

## 4. Development of the Distributed Monitoring System

According to the simulation results, the stress and strain of the transformer winding are mainly concentrated in the central region of the *l*_1_ and *l*_2_ paths under progressively short-circuit current shock impulse with an amplitude of 10–40 μm. There are slight differences in the vibration characteristics of different positions from the winding due to the relatively complex structure of the 3D wound core transformer, which puts higher requirements on the monitoring system. The framework of the designed monitoring system is shown in Figure 8, and the hardware of the system consists of several distributed wireless acceleration sensor nodes, a router, and a software monitoring terminal. The wireless sensor nodes encompass acceleration sensors, antennas, and WIFI wireless communication modules, which are mainly responsible for signal acquisition, processing, and communication. The router serves as the relay station of the local area network to manage the access of wireless node devices. The software system mainly fulfills the requirements of compressing signal data streams and displaying and storing data. All wireless sensor devices and terminal detection devices will be connected to the router to form an autonomous wireless local area network.

As the core component in the hardware system, the wireless sensor with high accuracy is required to accomplish data acquisition during short circuit conditions. In light of this, a piezoelectric PVDF vibration thin film sensor of cantilever beam structure (model LDT0-028K, TE Connectivity, Schaffhausen, Switzerland) with a wide frequency band (1 MHz–1000 MHz) and high sensitivity (~16 V/g) is selected. The designed wireless sensor node mainly includes a WIFI wireless communication module, antenna, micro-vibration sensor, measurement and control circuit, as well as the switching/charging module, as shown in Figure 9. The WIFI communication chip (model Quectel FC41D, Quectel Wireless Solutions Co., Ltd., Shanghai, China) has a chip processor frequency of up to 120 MHz, supporting the IEEE802.11b/g/n protocol, a built-in 256 KB RAM, and 4 MB Flash storage. Meanwhile, the WIFI chip has compact LCC packaging with dimensions of 20 mm × 18 mm × 2.6 mm, which can greatly satisfy the requirements of terminal products for miniaturized compact products and is compatible with diverse structural designs. The antenna adopts an omnidirectional high-gain FPC soft antenna with low loss. A stable signal is employed for easy installation and deployment on curved surfaces due to its flexible and bendable features. In view of the scattered distribution of the wireless sensor nodes, the power supply conditions are restricted and the wired power supply is not suitable in this situation. Therefore, lithium batteries are used to power the hardware system, and the CN3052A lithium battery charging management IC module is supplemented to manage its charging and discharging, as well as overheating protection. The packaging of wireless nodes mainly considers two factors: (i) Since the built-in FPC antenna needs to communicate with the outside world, the packaging material must be made of non-magnetic materials to avoid shielding electromagnetic waves. (ii) The instantaneous acceleration generated by a sudden short circuit of a transformer is about 100 times that of a steady-state operation, thus stable node installation and easy disassembly should be considered. Taking into account the above two points, the packaging shell is made of FR4 fiberglass material with a circular N50 magnet embedded in the low end for adsorption on the transformer. The shell is fixed with screws and reserved with charging and alarming light ports.

According to the above simulation analysis, the maximum force and deformation area of the high-voltage winding is evident near the midpoint area of the side path *l*_2_, while the maximum force and deformation area of the low-voltage winding is concentrated near the midpoint area of the front path *l*_1_. In this regard, six distributed wireless sensor nodes (DEV1#–DEV6#) are employed to monitor the vibration of the winding of the three-dimensional coil iron core transformer under progressively short-circuit conditions, as shown in Figure 10. To facilitate comparison and identification of vibration signals during short circuits, DEV1#–DEV3# were deployed on the individual phases of A, B, and C for the transformer for monitoring circumferential vibrations, while DEV4#–DEV5# were deployed on the top and bottom of the transformer for monitoring radial vibrations. DEV6# was deployed on the plane of the transformer common ground for monitoring the background vibration intensity during short circuit conditions. The coupling agent is evenly applied to the lower end of the permanent magnet of the sensor node device and adsorbed on the surface of the box. The collected data are transmitted to the monitoring terminal through a WIFI network.

For the software design, the C/S architecture of SOCKET communication was employed for the coil iron core transformer. The single-layer interactive C/S architecture ensures real-time data exchange within the substation due to its fast response. The specific design and implementation process of this software system is as follows: first, assign fixed IP addresses, port numbers, and device numbers online to wireless sensor nodes, routers, and servers, and build a small WIFI local area network in special scenarios. All intelligent devices such as transformer monitoring nodes operate in the same local area network, eliminating the need for wide-area network usage scenarios. The vibration frequency of the coil iron core power transformer reaches several tens of kHz once suddenly short-circuited, and the high sampling rate for data collection (50 kHz) results in a large amount of data. A total of 100 sets of data are compressed into a UDP message for one-time transmission, with a transmission interval of 5 ms. In this regard, the UDP protocol is used for data transmission. A connection between the client and server before transmitting data packets is not required in this protocol, and a single transmission with a fast response makes it more suitable for application scenarios with large amounts of data and fast response. The C# service program communicates with the client through a socket to receive data from the device port. In addition, placing the vast majority of business logic and interface display logic on the client program interface is more conducive to data analysis and processing, as well as graphical and rich interface display.

Now we compare the presented distributed wireless sensors monitoring system with similar state-of-the-art technologies for status monitoring of the distribution transformers. The use of the vibroacoustic method using a piezoelectric accelerometer was reported for core and winding measurement and condition assessment, and two criteria were proposed for assessing the condition of transformers based on the results of vibroacoustic tests [31]. Among versatile monitoring methods for the transformer conditions, optical measurement methods seem attractive due to their high spatial resolution, wider monitoring range, and high stabilities, and thus the distributed fiber optic sensing system is suitable for transformer monitoring [32,33,34]. Specifically, a viable method utilized square cantilever Fiber Bragg Grating (FBG) sensors with superior merit to suppress the ambient electromagnetic interferences, which were integrated into structures that increase the mechanical response of the overall system with desired sensitivities [33]. Another alternative of a quasi-distribution vibration sensing system for transformer online health monitoring based on phase-sensitive optical frequency domain reflectometry (OFDR) and a weak reflector sensing array is proposed, and in-lab and practical online monitoring tests for tank vibration were implemented on a 10 kV three-phase oil-immersed transformer [34]. Previously focused efforts were dedicated to monitoring the daily operation and maintenance of the online transformers. Even so, for the short-circuit tests of the grid admittance spot-check, related research mostly stays at the theoretical calculation and analysis for electromagnetic force and strain estimation caused by a short-circuit shock impulse [35,36]. In addition, most previous research subjects were mainly focused on the traditional stacked iron core distribution transformers. However, with the release of new energy efficiency standards, numerous high-energy efficient three-dimensional coil iron core distribution transformers will surge into the grid, posing a huge challenge to the efficient admittance spot-check of the transformers. Therefore, by contrast, the proposed distributed monitoring system has two differences in an application scenario and research subject relative to previous counterparts: (i) the proposed monitoring system is only applied to the admittance spot-check process of transformers for entering the grid, which is a beneficial supplement of the traditional short-circuit reactance detection methods and (ii) the detection subjects are mostly for efficient 3D wound core transformers.

For practical testing, the impulse current is applied with an approximately 80% short-circuit current shock impulse, and the six distributed sensor nodes were deployed on the exterior surface of the transformer for radial and axial shock acceleration monitoring. A single Phase A short circuit was implemented for this test, and then the phase voltage for A, B, and C terminals was applied to 15.35 kV, 14.25 kV, and 15.45 kV with the applied peak-value of the impulse current of 1149.9 A, 856 A, and 979.4 A. The overall impulse time lasts for approximately 216 ms. The captured waveforms during the shock impulse are shown in Figure 11. DEV1#–DEV3# were deployed to monitor the radial vibration corresponding to the three-core winding, respectively. Results show that the maximum shock acceleration reaches 3.64 m/s^2^ and lasts approximately 166 ms for DEV1# (Figure 11a). For DEV2#–DEV3#, the obtained maximum accelerated speed reached 3.7 m/s^2^ and 4.22 m/s^2^, which lasted 130 ms and 173 ms, respectively (Figure 11b,c). The axial vibration was monitored by DEV4# and DEV5# and deployed on the top and bottom of the transformer, respectively, and relatively lower intensities of shock accelerations were 2.5 m/s^2^ and 1.89 m/s^2^ with shorter lasting intervals. In addition, DEV1# was employed to monitor the background noise to exclude the influence of external interference. The vibrations fluctuate around 0.05 m/s^2^ and are immune from interferences, as shown in Figure 11f. We infer from the results that the winding deformation induced by a single-phase short-circuit shock impulse may occur in the location where DEV1# was deployed with maximum intensities and longer intervals. For further validating the observations, a subsequent reactance test was carried out with initial values for the A, B, and C terminals of 33.36 Ω, 32.39 Ω, and 31.32 Ω, respectively, and values of 33.59 Ω, 32.49 Ω, and 31.16 Ω in the event of a shock impulse. Obviously, the maximum reactance variation occurred at Phase A, which matches well with our prediction from the distributed monitoring system. Therefore, the presented distributed system can effectively detect the winding status during the short-circuit shock impulse, which is a beneficial supplement to the traditional reactance measurement method for efficient grid access permit testing.

## 5. Conclusions

In summary, a three-dimensional simulation model was established by taking the real three-phase three-winding transformer with the S20-MRL-400/10-NX2 (TGOOD Electric Co., Ltd., Qingdao, China) model as a reference. Taking advantage of the combination of the elastic–plastic large deflection analysis method and finite element analysis methods, the elastic–plastic deformation for the core windings of the three-phase three-dimensional coil iron core transformer under progressively sudden short circuit conditions is analyzed. Based on the estimated deformation distributions, a distributed monitoring system with integrated accelerometers was presented to capture the waveforms under short-circuit shock impulses. The results provide theoretical guidance for the design and quality spot-check of the three-dimensional coil iron core. Major findings were suggested below:

(1) Under the short-circuit shock impulses, the directions of equivalent stress and total deformation for the high/low-voltage windings are opposite to the asymmetric equivalent stress. The maximum force and deformation region for the high-voltage winding approaches the midpoint region of the side path *l*_2_, while the maximum force and deformation region for the low-voltage winding approaches the midpoint region of the front path *l*_1_.

(2) Under the condition of progressive shock impulses of applied short-circuit currents (60%, 80%, 100%, 120%), the variations in transformer short-circuit current will affect the distribution of equivalent stress and total deformation in the winding. Under the excitation of a 120% short-circuit current, the maximum equivalent stress and maximum total deformation reach 29.81 MPa and 38.70 MPa, respectively.

(3) The discrepancies in the distribution of equivalent stress and total deformation on the winding gradually increase with the applied short-circuit impulses, and the equivalent strain and total deformation on the winding are approximately proportional to the intensities of the short-circuit current. In addition, as the short-circuit current increases from 60% to 80%, the equivalent stress and total deformation variation rate reach their maximum of 177.75% and 177.43%, and 178.30% and 177.45%, respectively.

(4) Optimization deployment principles for wireless sensor nodes were proposed based on the simulation results, and a distributed monitoring system with WIFI networking, data acquisition, and data storage functions was developed to obtain the vibration status of the core windings under progressive short-circuited conditions. Practical testing was implemented with well-deployed distributed sensors under an 80% short-circuit shock impulse, and the observations were validated by the traditional reactance measurement method for an efficient grid access spot-check.

## Figures and Tables

**Figure 1 sensors-24-04062-f001:**
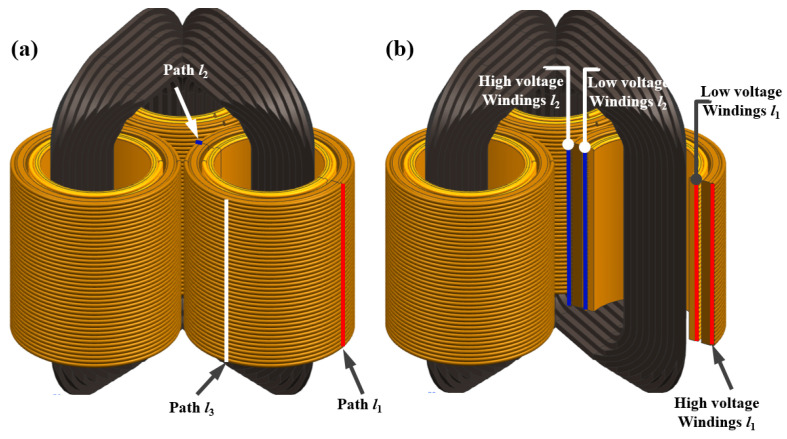
Schematic diagram of 3D wound core three-phase transformer (**a**,**b**) schematic diagram of data extraction path at the positions of high-voltage winding and low-voltage winding.

**Figure 2 sensors-24-04062-f002:**
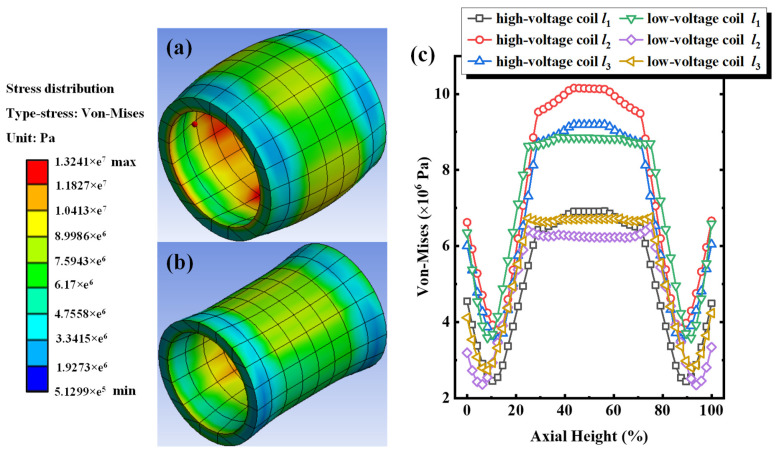
Under 80% short-circuit impulse for Phase A. (**a**) Cloud chart of equivalent stress distribution for high-voltage windings, (**b**) cloud chart of equivalent stress distribution for low-voltage windings, (**c**) equivalent stress along paths of *l*_1_, *l*_2_, and *l*_3_ for high/low-voltage windings.

**Figure 3 sensors-24-04062-f003:**
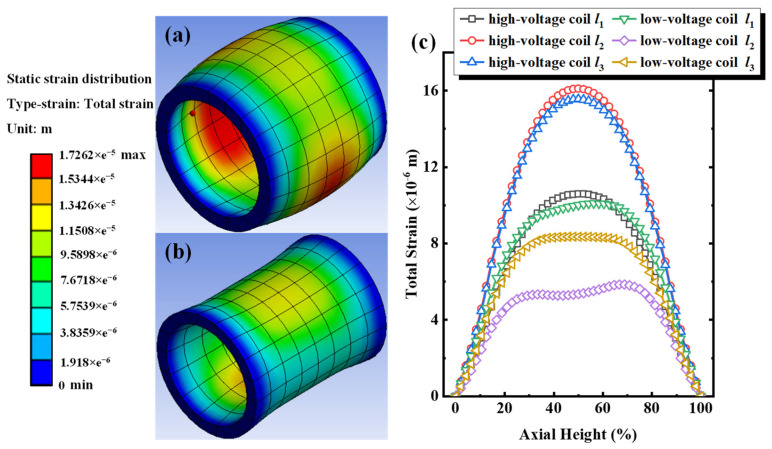
Under 80% short-circuit impulse for Phase A. (**a**) Cloud chart of equivalent deformation distribution for high-voltage windings, (**b**) cloud chart of equivalent deformation distribution for low-voltage windings, (**c**) equivalent deformation along paths of *l*_1_, *l*_2_, and *l*_3_ for high/low-voltage windings.

**Figure 4 sensors-24-04062-f004:**
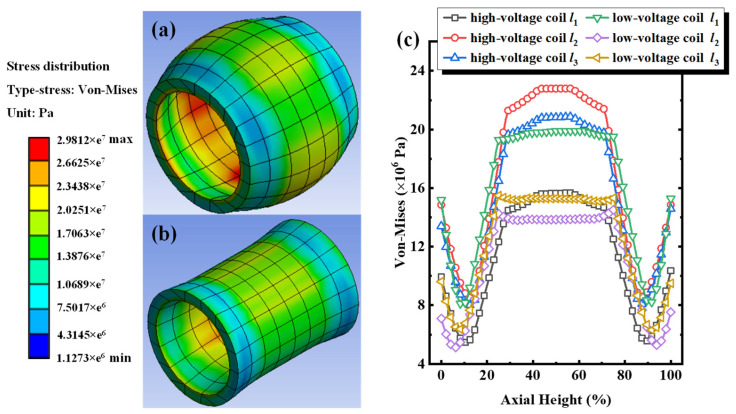
Under 120% short-circuit impulse for Phase A. (**a**) Cloud chart of equivalent stress distribution for high-voltage windings, (**b**) cloud chart of equivalent stress distribution for low-voltage windings, (**c**) equivalent stress along paths of *l*_1_, *l*_2_, and *l*_3_ for high/low-voltage winding.

**Figure 5 sensors-24-04062-f005:**
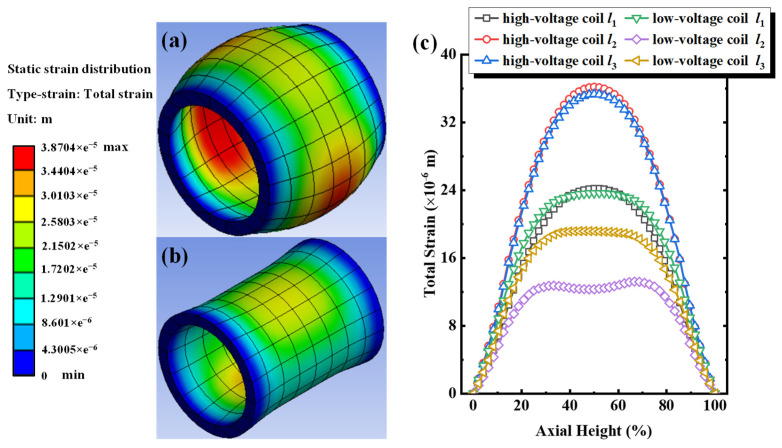
Under 120% short-circuit impulse for Phase A. (**a**) Cloud chart of equivalent deformation distribution for high-voltage windings, (**b**) cloud chart of equivalent deformation distribution for low-voltage windings, (**c**) equivalent deformation along paths of *l*_1_, *l*_2_, and *l*_3_ for high/low-voltage windings.

**Figure 6 sensors-24-04062-f006:**
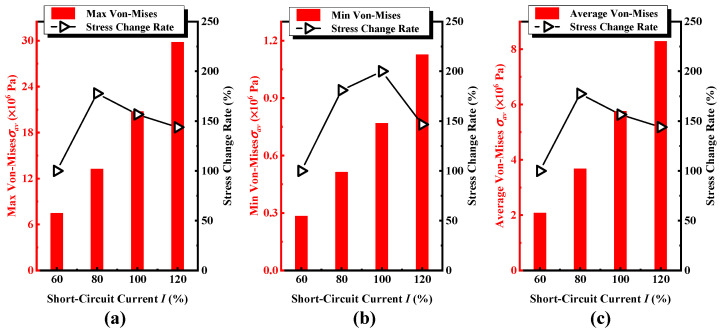
Under cumulative short-circuit impulse currents (**a**) the maximum equivalent (Von Mises) stress and its change rate in the A-phase winding, (**b**) the minimum equivalent (Von Mises) stress and its change rate, (**c**) the average equivalent (Von Mises) stress and its change rate.

**Figure 7 sensors-24-04062-f007:**
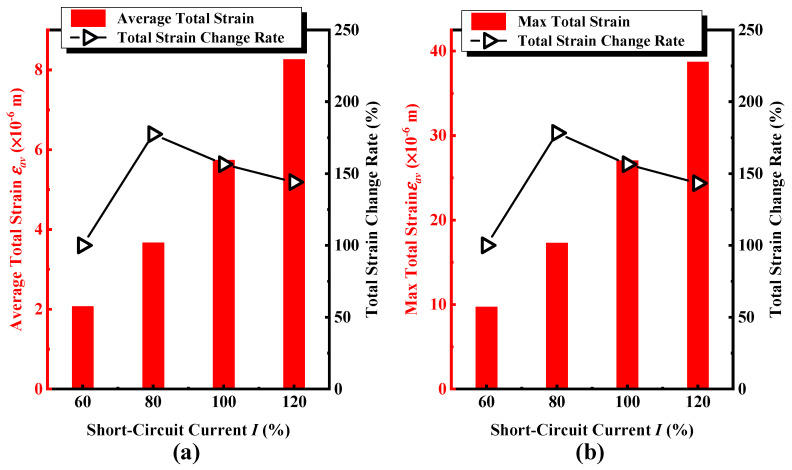
Under cumulative short-circuit impulse currents (**a**) the maximum equivalent (Von Mises) deformation and its change rate in the A-phase winding, (**b**) the minimum equivalent (Von Mises) deformation and its change rate.

**Figure 8 sensors-24-04062-f008:**
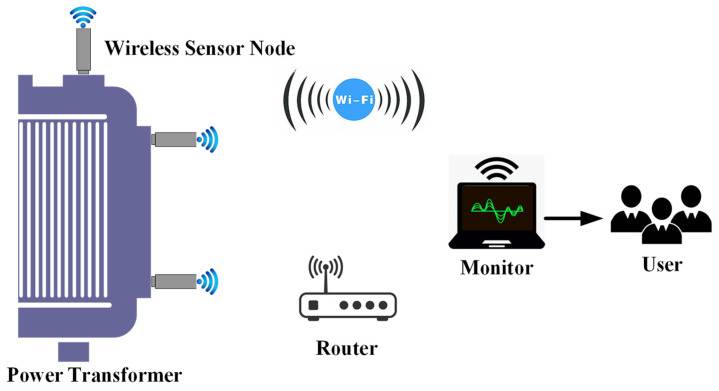
Schematic diagram of distributed monitoring system.

**Figure 9 sensors-24-04062-f009:**
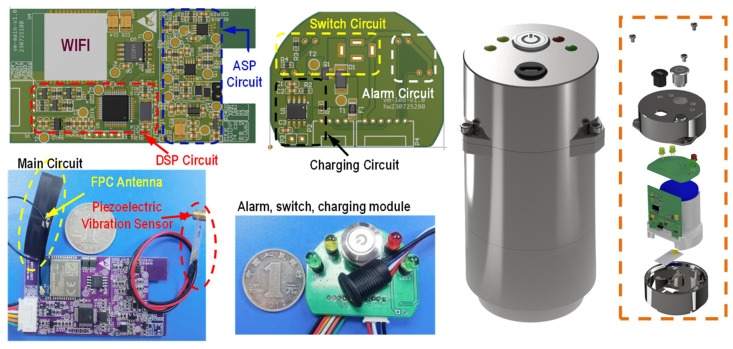
Hardware and packaging design of the wireless sensor nodes.

**Figure 10 sensors-24-04062-f010:**
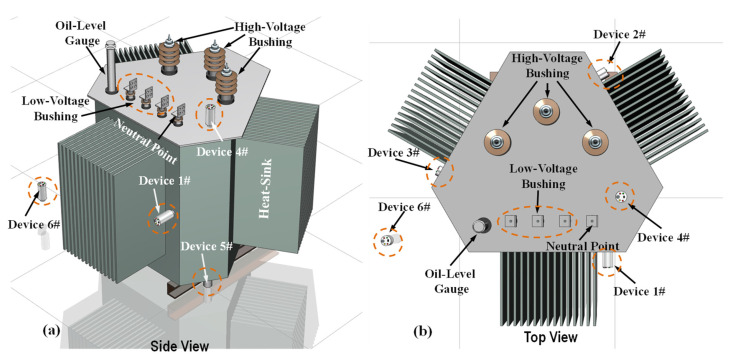
Schematic diagram of deployment principles for wireless sensor nodes: (**a**) side view of the deployed monitoring sensor nodes, (**b**) top view of the deployed monitoring sensor nodes.

**Figure 11 sensors-24-04062-f011:**
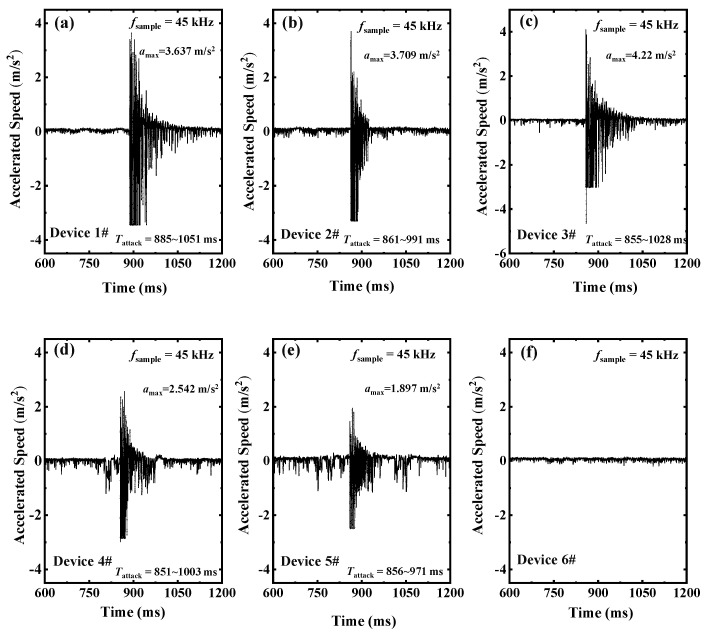
Captured waveforms for the distributed accelerometers from (**a**) Device 1#, (**b**) Device 2#, (**c**) Device 3#, (**d**) Device 4#, (**e**) Device 5#, (**f**) Device 6#, under 80% short circuit impulse for single phase.

**Table 1 sensors-24-04062-t001:** Foundational parameters of transformers.

Structural Component	Material	Conductivity/S/m	Permeability
Core	Fe-based amorphous alloy	2.42 × 10^6^	B-H curve
Coil	GB/T 7095.2 [30]	4.5 × 10^7^	/
Oil tank	4.0 Steel plate Q235B	6.484 × 10^6^	B-H curve
Steel plate	6.0 Steel plate Q235B	6.484 × 10^6^	B-H curve
Magnetic screen	30RGH120	Anisotropic	Anisotropic
Pull plate (non-steel magnetic)	20Mn23AL	1.3889 × 10^6^	/

## Data Availability

Data will be made available upon reasonable request.

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
