# Peer review of "Distributed Vibration Monitoring System for 10 kV-400 kVA 3D Wound Core Transformer under Progressive Short-Circuit Impulses"

_sensors, 2024, doi:10.3390/s24134062_

Round 1
Reviewer 1 Report
Comments and Suggestions for Authors
This paper introduces a distributed vibration monitoring system for transformers under progressive short-circuit impulses. The work is interesting. However, there are some issues that should be addressed before this paper can be accepted. The detailed comments are as follows:
1. In the introduction, the authors should clearly summarize their technical contributions. Moreover, the literature review is not comprehensive enough, and some recent literature can be considered, such as 10.1016/j.engappai.2023.107382, or 10.1109/JSEN.2024.3395970.
2. The work described in the paper is rich, but some contents can be more concise, or they can be shown in the form of figures.
3. There is not enough discussion about the testing and validation of the proposed system.
4. It will be more convincing if the proposed system is compared with similar existing systems.
5. The English writing of the paper needs improvement. The authors can seek the help of native English speakers to edit the paper.
Comments on the Quality of English LanguageModerate editing of English language is required.
Reviewer 2 Report
Comments and Suggestions for Authors
1. The authors should maintain a constant gap spacing between the magnitude and unit (Eg. 29.81 MPa) throughout the manuscript.
2. Does the transmission of data via WIFI for the current waveforms during short circuit conditions impact the actual data in the transformer?
3. Why the strain increased for the short circuit percentage of 60% to 80% whereas it decreases after this level? Explain.
Reviewer 3 Report
Comments and Suggestions for Authors
The authors propose a three-dimensional model of a three-phase, three-winding transformer, in which several conditions are studied, and the principles for wireless sensor nodes are proposed.
Some comments:
a) The absence of noise in the proposed simulation is a significant concern. It's crucial to consider two types of uncertainties in simulations: measurement noise (Gaussian) and system noise from unidentified sources. By incorporating these noise considerations into simulations, the authors can potentially enhance the performance of the experiment and the validity of the proposed methodology. The authors should clearly justify their choice not to consider the noise.
b) Another issue concerns the application of your proposed methodology to real transformers. A comparison should be made to validate the proposed simulation model.
Round 2
Reviewer 1 Report
Comments and Suggestions for Authors
The authors have revised their manuscript carefully, and all my comments have been responded to appropriately. Therefore, the manuscript can be accepted now.
Comments on the Quality of English LanguageMinor editing of English language is required.
Reviewer 3 Report
Comments and Suggestions for Authors
After revising this new manuscript and the author’s responses, this reviewer can conclude that the new submission has significantly improved from the first version, and my concerns have been clarified adequately.